# INVASE: INSTANCE-WISE VARIABLE SELECTION USING NEURAL NETWORKS

**Jinsung Yoon**
Department of Electrical and Computer Engineering
UCLA, California, USA
jsyoon0823@g.ucla.edu

**James Jordon**
Engineering Science Department
University of Oxford, UK
james.jordon@wolfson.ox.ac.uk

**Mihaela van der Schaar**
University of Cambridge, UK
Department of Electrical and Computer Engineering, UCLA, California, USA
Alan Turing Institute, London, UK
mihaela@ee.ucla.edu

## ABSTRACT

The advent of big data brings with it data with more and more dimensions and thus a growing need to be able to efficiently select which features to use for a variety of problems. While global feature selection has been a well-studied problem for quite some time, only recently has the paradigm of instance-wise feature selection been developed. In this paper, we propose a new instance-wise feature selection method, which we term INVASE. INVASE consists of 3 neural networks, a selector network, a predictor network and a baseline network which are used to train the selector network using the actor-critic methodology. Using this methodology, INVASE is capable of flexibly discovering feature subsets of a different size for each instance, which is a key limitation of existing state-of-the-art methods. We demonstrate through a mixture of synthetic and real data experiments that INVASE significantly outperforms state-of-the-art benchmarks.

## 1 INTRODUCTION

High-dimensional data is becoming more readily available, and it brings with it a growing need to be able to efficiently select which features to use for a variety of problems. When doing predictions, it is well known that using too many variables with too few samples can lead to overfitting, which can significantly hinder the performance of predictive models. In the realm of interpretability, the large dimensionality of the data is often too much information to present to a human who may be using the machine learning model as a support system. Understanding which features are most relevant to an outcome or to a model output is an important first step in improving predictions and interpretability and many works exist that tackle feature selection on a global level. However, in the heterogeneous data we typically encounter, the prediction made by a model (and indeed the true label) may rely on a different subset of the features for different subgroups within the data [14]. In this paper we propose a novel *instance-wise* feature selection method, INVASE (INstance-wise VAriable SElection), which attempts to learn which subset of the features is relevant for each sample, allowing us to display the minimal information required to explain each prediction and also to reduce overfitting of predictive models.

Discovering a global subset of relevant features for a particular task is a well-studied problem and there are several existing methods for solving it such as Sequential Correlation Feature Selection [11], Mutual Information Feature Selection [21], Knockoff models [3], and more [10; 16]. However, global feature selection suffers from a key limitation - the features discovered by global feature selection are the same for all samples. In many cases, in particular when populations are highly heterogeneous, the relevant features may differ across samples [33; 32]. For instance, different patient subgroups have different relevant features for predicting heart failure [14]. *Instance-wise* feature selection methods such as [4; 27] instead try to discover the features that are relevant *for*

*each sample*. When the goal is to provide an interpretable explanation of the predictions made, a key challenge is in ensuring that we do not *over-explain* by providing too much information (i.e. choosing too many features). Naturally, by performing feature selection on an individualized level we are able to select features that are more relevant to each sample, rather than having to choose the top $k$ features globally, which may not explain the predictions for some samples very well, but simply perform well on average across all samples.

In this paper, we propose a novel instance-wise feature selection method which we term INVASE. We draw influence from actor-critic models [22] to solve the problem of backpropagating through subset sampling. Our model consists of 3 neural networks: a selector network, a predictor network and a baseline network. During training, each of these are trained iteratively, with the selector network being trained to minimize a Kullback-Leibler (KL) divergence between the full conditional distribution and the selected-features-only conditional distribution of the outcome. Our model is capable of discovering a different number of relevant variables for each sample which is a key limitation in existing instance-wise approaches (such as [4]). We show significant improvements over the state-of-the-art in both synthetic data and real-word data in terms of true positive rates, false discovery rates, and show better predictive performance with respect to several prediction metrics. Our model can also be easily extended to handle both continuous and discrete outputs and time-series inputs (see the Appendix for details).

## 1.1 RELATED WORKS

There are many existing works on global variable selection (see [10] for a good summary paper). [21] and [11] use max-dependency min-redundancy criteria [17] with mutual information and Pearson correlation, respectively. [3] uses multiple hypothesis testing for global variable selection. As noted above, these global selection methods are not capable of learning sample-specific relevance.

Instance-wise variable selection is also closely related to model interpretation methods. Some previous works are based on backpropagation from the output of the predictive model to the input variables [29]. DeepLIFT [27] decomposes the output of the neural network on a reference input to compute the contribution of each input variable. However, both methods need white-box access to the pre-trained predictive models to compute the gradient and decomposition. [2] approximates the predictive models using a Parzen window approximator when there is only black-box access to the predictive models. Some other works are based on input perturbation such as [1], [15], [30] and [5]. [18] uses Shapley values to compute the variable importance, and [24] uses locally linear models to explain the linear dependency for each sample. [19] tries to interpret tree ensemble models using Shapley values but cannot generalize to other predictive models such as neural networks.

Our work is most closely related to L2X (Learning to Explain) [4]. However, there are 3 key differences between our work and theirs. In L2X, they try to maximize a lower bound of the mutual information between the target $Y$ and the selected input variables $X_S$. In contrast, we try to minimize the KL divergence between the conditional distributions $Y|X$ and $Y|X_S$. In order to be able to backpropagate through subset sampling, L2X use the Gumbel-softmax trick [13] to approximately discretize the continuous outputs of the neural network. In our work, we use methods from actor-critic models [22] to bypass backpropagation through the sampling and instead use the predictor network to provide a reward to the selector network. Finally, due to the Gumbel-softmax used in L2X, the number of variables to be detected must be fixed in advance and is necessarily the same for every sample. The actor-critic methodology used in our model has no such limitations and so we are able to flexibly select a different number of relevant variables for each sample and instead induce sparsity using an $l_0$ penalty term. In fact, using the actor-critic methodology allows us to directly use the $l_0$ penalty term (which is not differentiable and therefore not practical to use in general). A summary table highlighting the key features of all of the related works can be found in the Appendix.

## 2 PROBLEM FORMULATION

Let $\mathcal{X} = \mathcal{X}_1 \times ... \times \mathcal{X}_d$ be a $d$-dimensional feature space and $\mathcal{Y} = \{1, ..., c\}$ be a discrete label space[1]. Let $\mathbf{X} = (X_1, ..., X_d) \in \mathcal{X}$ and $Y \in \mathcal{Y}$ be random variables with joint density (or mass) $p$ and marginal densities (or masses) $p_X$ and $p_Y$ respectively. We will refer to $\mathbf{s} \in \{0, 1\}^d$ as the

---

[1]In this paper we focus on classification; we discuss an extension of our model to regression in the Appendix.

selection vector, where $s_i = 1$ will indicate that variable $i$ is selected, and $s_i = 0$ will indicate that variable $i$ is not selected. Let $*$ be any point not in any of the spaces $\mathcal{X}_1, ..., \mathcal{X}_d$ and define $\mathcal{X}_i^* = \mathcal{X}_i \cup \{*\}$ and $\mathcal{X}^* = \mathcal{X}_1^* \times ... \times \mathcal{X}_d^*$. Given $\mathbf{x} \in \mathcal{X}$ we will write $\mathbf{x}^{(\mathbf{s})}$ to denote the suppressed feature vector defined by

$$x_i^{(\mathbf{s})} = \begin{cases} x_i \text{ if } s_i = 1 \\ * \text{ if } s_i = 0 \end{cases}$$

so that $*$ represents that a feature is not selected.

In the global feature selection literature, the goal is to find the smallest $\mathbf{s}$ (i.e. the one with fewest 1s) such that $\mathbb{E}(Y|\mathbf{X}^{(\mathbf{s})}) = \mathbb{E}(Y|\mathbf{X})$, or equivalently such that the conditional distribution of $Y$ given $\mathbf{X}^{(\mathbf{s})}$ is the same as $Y$ given all of $\mathbf{X}$. Note that this definition is given fully in terms of random variables, rather than realizations of those random variables.

In contrast, our problem necessarily needs to be defined in terms of realizations since we are aiming to select features *for a given realization*. We will write $\mathbf{x}$ to denote realizations of the random variable $\mathbf{X}$. Then we formalize our problem as one of finding a selector function, $S : \mathcal{X} \to \{0,1\}^d$ such that for almost every $\mathbf{x} \in \mathcal{X}$ (w.r.t. $p_X$) we have

$$(Y|\mathbf{X}^{(S(\mathbf{x}))} = \mathbf{x}^{(S(\mathbf{x}))}) \overset{d.}{=} (Y|\mathbf{X} = \mathbf{x}) \tag{1}$$

where $\overset{d.}{=}$ denotes equality in distribution and $S(\mathbf{x})$ is minimal (i.e. fewest 1s) such that (1) holds.

We suppose that we have a dataset $\mathcal{D} = \{(\mathbf{x}_j, y_j)\}_{j=1}^n$ consisting of $n$ i.i.d. realizations of the pair $(\mathbf{X}, Y)$.[2] Note that $Y$ can be viewed as having either come from a dataset, in which case the problem is of selecting predictive features, or as having come from a predictive model, in which case the problem is of explaining the model's predictions.

## 2.1 OPTIMIZATION PROBLEM

In order to learn a suitable selector function, we transform the constraint (1) into a soft constraint using the Kullback-Leibler (KL) divergence which, for random variables $W$ and $V$ with densities $p_W$ and $p_V$ is defined as

$$KL(W||V) = \mathbb{E}\left[\log\left(\frac{p_W(W)}{p_V(W)}\right)\right].$$

We define the following loss for our selector function $S$

$$\mathcal{L}(S) = \mathbb{E}_{\mathbf{x} \sim p_X}\left[KL(Y|\mathbf{X} = \mathbf{x}||Y|\mathbf{X}^{(S(\mathbf{x}))} = \mathbf{x}^{(S(\mathbf{x}))}) + \lambda||S(\mathbf{x})||\right] \tag{2}$$

where $|| \cdot ||$ simply denotes the number of non-zero entries of a vector (or equivalently in this case, the number of 1s) and $\lambda$ is a hyper-parameter that trades off between the constraint in (1) and the number of selected features. The KL divergence in (2) can be rewritten as

$$KL(Y|\mathbf{X} = \mathbf{x}||Y|\mathbf{X}^{(S(\mathbf{x}))} = \mathbf{x}^{(S(\mathbf{x}))}) = \mathbb{E}_{y \sim Y|\mathbf{X} = \mathbf{x}}\left[\log\left(\frac{p_Y(y|\mathbf{x})}{p_Y(y|\mathbf{x}^{(S(\mathbf{x}))})}\right)\right]$$

$$= \mathbb{E}_{y \sim Y|\mathbf{X} = \mathbf{x}}\left[\log(p_Y(y|\mathbf{x})) - \log(p_Y(y|\mathbf{x}^{(S(\mathbf{x}))}))\right]$$

$$= \int_{\mathcal{Y}} p_Y(y|\mathbf{x})\left[\log(p_Y(y|\mathbf{x})) - \log(p_Y(y|\mathbf{x}^{(S(\mathbf{x}))}))\right] dy$$

where $p_Y(\cdot|\cdot)$ denotes the appropriate conditional densities of $Y$. We will write

$$l(\mathbf{x}, \mathbf{s}) = \int_{\mathcal{Y}} p_Y(y|\mathbf{x})\left[\log(p_Y(y|\mathbf{x})) - \log(p_Y(y|\mathbf{x}^{(s)}))\right] dy \tag{3}$$

so that our final loss can be written as

$$\mathcal{L}(S) = \mathbb{E}_{\mathbf{x} \sim p_X}\left[l(\mathbf{x}, S(\mathbf{x})) + \lambda||S(\mathbf{x})||\right] \tag{4}$$

where $|| \cdot ||$ denotes the $l_0$ (pseudo-)norm.

---

[2]We will occasionally abuse notation and write $y_i$ to denote the $i$th element of the one-hot encoding of $y$, though the context should make it clear when this is the case.

## 3    PROPOSED MODEL

There are two main challenges in minimizing the loss in (4). First, the output space of the selector function ($\{0, 1\}^d$) is large - its size increases exponentially with the dimension of the feature space; thus a complete search is impractical in high dimensional settings (and it should be noted that it is in high dimensional settings where feature selection is most necessary). Second, we do not have access to the densities $p_Y(\cdot|\mathbf{x}^{(S(\mathbf{x}))})$ and $p_Y(y|\mathbf{x})$ required to compute (4).

### 3.1    LOSS ESTIMATION

To approximate the densities in (3), we introduce a pair of functions $f^\phi : \mathcal{X}^* \times \{0, 1\}^d \to [0, 1]^c$ parametrized by $\phi$ and $f^\gamma : \mathcal{X} \to [0, 1]^c$ parametrized by $\gamma$ that will estimate $p_Y(\cdot|\mathbf{x}^{(S(\mathbf{x}))})$ and $p_Y(\cdot|\mathbf{x})$ respectively.

#### 3.1.1    PREDICTOR NETWORK

We refer to $f^\phi$ as the predictor network. This will take as input a suppressed[3] feature vector $\mathbf{x}^{(\mathbf{s})}$ and its corresponding selection vector $\mathbf{s}$ and will output a probability distribution (using a softmax layer) over the $c$-dimensional output space.

$f^\phi$ is trained to minimize the cross entropy loss given by

$$l_1(\phi) = -\mathbb{E}_{(\mathbf{x},y) \sim p, \mathbf{s} \sim \pi_\theta(\mathbf{x}, \cdot)} \left[ \sum_{i=1}^c y_i \log(f_i^\phi(\mathbf{x}^{(\mathbf{s})}, \mathbf{s})) \right]$$

where $y_i$ is the $i$th component of the one-hot encoding of $y$ and $\pi_\theta$ is the distribution induced by our selector network which will be defined in the following section. $f^\phi$ is implemented as a fully connected neural network[4].

#### 3.1.2    BASELINE NETWORK

We refer to $f^\gamma$ as the baseline network, which is standard in the actor-critic literature for variance reduction. $f^\gamma$ is implemented as a fully connected neural network and is trained to minimize

$$l_3(\gamma) = -\mathbb{E}_{(\mathbf{x},y) \sim p} \left[ \sum_{i=1}^c y_i \log(f_i^\gamma(\mathbf{x})) \right].$$

For fixed $\phi, \gamma$ we define our loss estimator, $\hat{l}$, by

$$\hat{l}(\mathbf{x}, \mathbf{s}) = - \left[ \sum_{i=1}^c y_i \log(f_i^\phi(\mathbf{x}^{(\mathbf{s})}, \mathbf{s})) - \sum_{i=1}^c y_i \log(f_i^\gamma(\mathbf{x})) \right]. \tag{5}$$

### 3.2    SELECTOR FUNCTION OPTIMIZATION

We approximate the selector function $S : \mathcal{X} \to \{0, 1\}^d$ by using a single neural network, $\hat{S}^\theta : \mathcal{X} \to [0, 1]^d$ parameterized by weights $\theta$, that outputs a probability for selecting each feature (i.e. the $i$th component of $\hat{S}^\theta(\mathbf{x})$ will denote the probability with which we select the $i$th feature). The selector network induces a probability distribution over the selection space ($\{0, 1\}^d$), with the probability of a given joint selection vector $\mathbf{s} \in \{0, 1\}^d$ being given by[5]

$$\pi_\theta(\mathbf{x}, \mathbf{s}) = \Pi_{i=1}^d \hat{S}_i^\theta(\mathbf{x})^{s_i} (1 - \hat{S}_i^\theta(\mathbf{x}))^{1-s_i}.$$

---

[3]When implemented we set $* = 0$ and include the selection vector to differentiate this from the case $x_i = 0$.

[4]$f^\phi$, $f^\gamma$ and $\hat{S}^\theta$ could also be implemented as CNNs or RNNs, when appropriate.

[5]Note that, when $d$ is large, this becomes vanishingly small, however, $\pi_\theta$ appears in our loss only via its log and so in practice this is not a problem.

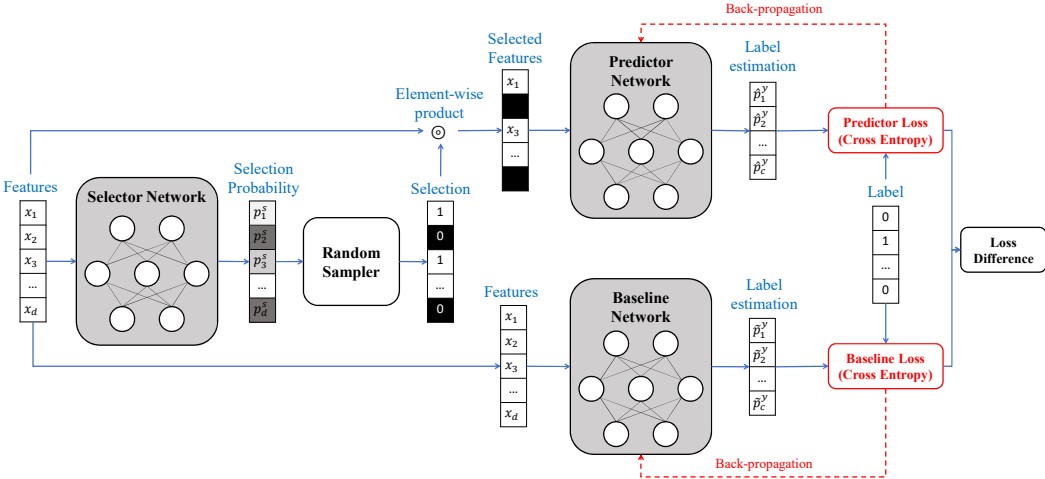

Figure 1: **Block diagram of INVASE.** Instances are fed into the selector network which outputs a vector of selection probabilities. The selection vector is then sampled according to these probabilities. The predictor network then receives the selected features and makes a prediction and the baseline network is given the entire feature vector and makes a prediction. Each of these networks are trained using backpropagation using the real label. The loss of the baseline network is then subtracted from the prediction network's loss and this is used to update the selector network.

Using this, we define the following loss for our selector network

$$l_2(\theta) = \mathbb{E}_{(\mathbf{x},y)\sim p}\left[\mathbb{E}_{\mathbf{s}\sim\pi_\theta(\mathbf{x},\cdot)}\left[\hat{l}(\mathbf{x},\mathbf{s}) + \lambda||\mathbf{s}||_0\right]\right]$$

$$= \int_{\mathcal{X}\times\mathcal{Y}} p(\mathbf{x},y)\left(\sum_{\mathbf{s}\in\{0,1\}^d}\pi_\theta(\mathbf{x},\mathbf{s})\left(\hat{l}(\mathbf{x},\mathbf{s}) + \lambda||\mathbf{s}||_0\right)\right)dxdy.$$

Taking the gradient of this loss with respect to $\theta$ gives us

$$\nabla_\theta l_2(\theta) = \int_{\mathcal{X}\times\mathcal{Y}} p(\mathbf{x},y)\left(\sum_{\mathbf{s}\in\{0,1\}^d}\nabla_\theta\pi_\theta(\mathbf{x},\mathbf{s})\left(\hat{l}(\mathbf{x},\mathbf{s}) + \lambda||\mathbf{s}||_0\right)\right)dxdy$$

$$= \int_{\mathcal{X}\times\mathcal{Y}} p(\mathbf{x},y)\left(\sum_{\mathbf{s}\in\{0,1\}^d}\frac{\nabla_\theta\pi_\theta(\mathbf{x},\mathbf{s})}{\pi_\theta(\mathbf{x},\mathbf{s})}\pi_\theta(\mathbf{x},\mathbf{s})\left(\hat{l}(\mathbf{x},\mathbf{s}) + \lambda||\mathbf{s}||_0\right)\right)dxdy$$

$$= \int_{\mathcal{X}\times\mathcal{Y}} p(\mathbf{x},y)\left(\sum_{\mathbf{s}\in\{0,1\}^d}\nabla_\theta\log\pi_\theta(\mathbf{x},\mathbf{s})\pi_\theta(\mathbf{x},\mathbf{s})\left(\hat{l}(\mathbf{x},\mathbf{s}) + \lambda||\mathbf{s}||_0\right)\right)dxdy$$

$$= \mathbb{E}_{(\mathbf{x},y)\sim p}\left[\mathbb{E}_{\mathbf{s}\sim\pi_\theta(\mathbf{x},\cdot)}\left[\left(\hat{l}(\mathbf{x},\mathbf{s}) + \lambda||\mathbf{s}||_0\right)\nabla_\theta\log\pi_\theta(\mathbf{x},\mathbf{s})\right]\right].$$

We update each of $\hat{S}^\theta$, $f^\phi$ and $f^\gamma$ iteratively using stochastic gradient descent. Pseudo-code of INVASE is given in Algorithm 1 and a block representation of INVASE can be found in Fig. 1.

## 4  EXPERIMENTS

In this section, we quantitatively evaluate INVASE against various state-of-the-art benchmarks on both synthetic and real-world datasets. We evaluate our performance both at identifying ground truth relevance and at enhancing predictions. We compare our model with 4 global variable selection models: Knockoffs [3], Tree Ensembles (Tree) [7], Sequential Correlation Feature Selection (SCFS) [11], and LASSO regularized linear model; and 3 instance-wise feature selection methods: L2X

---

**Algorithm 1** Pseudo-code of INVASE

1: **Inputs:** learning rates $\alpha, \beta > 0$, mini-batch size $n_{mb} > 0$, dataset $\mathcal{D}$
2: **Initialize** parameters $\theta, \phi, \gamma$
3: **while** Converge **do**
4:     Sample a mini-batch from the dataset $(\mathbf{x}_j, y_j)_{j=1}^{n_{mb}} \sim \mathcal{D}$
5:     **for** $j = 1, ..., n_{mb}$ **do**
6:         Calculate selection probabilities

$$(p_1^j, ..., p_d^j) \leftarrow \hat{S}^\theta(\mathbf{x}_j)$$

7:         Sample selection vector
8:         **for** $i = 1, ..., d$ **do**

$$s_i^j \sim Ber(p_i^j)$$

9:         Calculate loss

$$\hat{l}_j(\mathbf{x}_j, \mathbf{s}_j) \leftarrow - \left[ \sum_{i=1}^{c} y_i^j \log(f_i^\phi(\mathbf{x}_j^{(\mathbf{s}_j)}, \mathbf{s}_j)) - \sum_{i=1}^{c} y_i^j \log(f_i^\gamma(\mathbf{x}_j)) \right]$$

10:     Update the selector network parameters $\theta$

$$\theta \leftarrow \theta - \alpha \frac{1}{n_{mb}} \sum_{j=1}^{n_{mb}} \left( \hat{l}_j(\mathbf{x}_j, \mathbf{s}_j) + \lambda ||\mathbf{s}_j|| \right) \nabla_\theta \log \pi_\theta(\mathbf{x}_j, \mathbf{s}_j)$$

11:     Update the predictor network parameters $\phi$

$$\phi \leftarrow \phi - \beta \frac{1}{n_{mb}} \sum_{j=1}^{n_{mb}} \sum_{i=1}^{c} y_i^j \times \nabla_\phi \log(f_i^\phi(\mathbf{x}_j^{(\mathbf{s}_j)}, \mathbf{s}_j))$$

12:     Update the baseline network parameters $\gamma$

$$\gamma \leftarrow \gamma - \beta \frac{1}{n_{mb}} \sum_{j=1}^{n_{mb}} \sum_{i=1}^{c} y_i^j \times \nabla_\gamma \log(f_i^\gamma(\mathbf{x}_j))$$

---

[4], LIME [24], and Shapley [18]. The details of benchmark implementation can be found in the appendix. Implementation of INVASE can be found at `https://github.com/jsyoon0823/INVASE`.

### 4.1 SYNTHETIC DATA EXPERIMENTS

#### 4.1.1 EXPERIMENTAL SETTINGS

For our first set of experiments, we use the same synthetic data generation models as in L2X [4]. The input features are generated from an 11-dimensional[6][7] Gaussian distribution with no correlations across the features ($\mathbf{X} \sim \mathcal{N}(\mathbf{0}, \mathbf{I})$). The label $Y$ is sampled as a Bernoulli random variable with $\mathbb{P}(Y = 1|\mathbf{X}) = \frac{1}{1+\text{logit}(\mathbf{X})}$, where $\text{logit}(\mathbf{X})$ is varied to create 3 different synthetic datasets:

- **Syn1:** $\exp(X_1 X_2)$

- **Syn2:** $\exp(\sum_{i=3}^{6} X_i^2 - 4)$

- **Syn3:** $-10 \times \sin 2X_7 + 2|X_8| + X_9 + \exp(-X_{10})$

---

[6]In L2X they use a 10-dimensional Gaussian, we introduce $X_{11}$ to act as a "switch" to create instance-wise relevance. Experiments where instead the "switch" variable is one of $X_1, ..., X_{10}$ can be found in the appendix.

[7]We also perform experiments using 100 features in the Appendix to demonstrate the scalability of our method.

In each of these datasets, the label depends on the same subset of features for every sample. To highlight the capability of INVASE to detect instance-wise dependence, we generate 3 further synthetic datasets as follows:

- **Syn4:** If $X_{11} < 0$, logit follows **Syn1**, otherwise, logit follows **Syn2**.
- **Syn5:** If $X_{11} < 0$, logit follows **Syn1**, otherwise, logit follows **Syn3**.
- **Syn6:** If $X_{11} < 0$, logit follows **Syn2**, otherwise, logit follows **Syn3**.

Note that in **Syn4** and **Syn5**, the number of relevant features is different for different samples.

For each of **Syn1** to **Syn6** we draw 20,000 samples from the data generation model and separate each into training ($\mathcal{D}_{train} = (\mathbf{x}_i, y_i)_{i=1}^{10000}$) and testing ($\mathcal{D}_{test} = (\mathbf{x}_j, y_j)_{j=1}^{10000}$) sets. For each method we try to find the top $k$ relevant features for each sample (we set $k = 4$ for **Syn1, Syn2, Syn3, Syn4, Syn5** and $k = 5$ for **Syn6**), note, however, that $k$ is not given as an input to INVASE (but is necessary for other methods). The performance metrics we use are the true positive rate (TPR) (higher is better) and false discovery rate (FDR)[8] (lower is better) to measure the performance of the methods when the focus is on discovery (i.e. discovering which features are relevant) and we use Area Under the Receiver Operating Characteristic Curve (AUROC), Area Under the Precision Recall Curve (AUPRC) and accuracy when the focus is on predictions.

### 4.1.2 DISCOVERY

| Dataset | Syn1 | | Syn2 | | Syn3 | | Syn4 | | Syn5 | | Syn6 | |
|---|---|---|---|---|---|---|---|---|---|---|---|---|
| Metrics (%) | TPR | FDR | TPR | FDR | TPR | FDR | TPR | FDR | TPR | FDR | TPR | FDR |
| **INVASE** | **100.0** | **0.0** | **100.0** | **0.0** | **92.0** | **0.0** | **99.8** | **10.3** | **84.8** | **1.1** | **90.1** | **7.4** |
| L2X | 100.0 | 0.0 | 100.0 | 0.0 | 69.4 | 30.6 | 79.5 | 21.8 | 74.8 | 26.3 | 83.3 | 16.7 |
| LIME | 13.8 | 86.2 | 100.0 | 0.0 | 98.1 | 1.9 | 40.7 | 49.4 | 41.1 | 50.6 | 50.5 | 49.5 |
| Shapley | 60.4 | 39.6 | 93.3 | 6.7 | 90.9 | 9.1 | 65.2 | 31.9 | 62.9 | 33.7 | 71.2 | 28.8 |
| Knockoff | 10.0 | 70.0 | 8.7 | 36.2 | 81.2 | 17.5 | 38.8 | 35.1 | 41.0 | 51.1 | 56.6 | 42.1 |
| Tree | 100.0 | 0.0 | 100.0 | 0.0 | 100.0 | 0.0 | 54.7 | 39.0 | 56.8 | 37.5 | 60.0 | 40.0 |
| SCFS | 23.5 | 76.5 | 39.5 | 60.5 | 78.3 | 22.0 | 48.9 | 52.4 | 42.4 | 51.2 | 56.1 | 43.9 |
| LASSO | 19.0 | 81.0 | 39.8 | 60.2 | 78.3 | 21.7 | 49.9 | 50.9 | 45.5 | 48.2 | 56.4 | 43.6 |

Table 1: Relevant feature discovery results for Synthetic datasets with 11 features

As demonstrated by Table 1, our method is capable of detecting relevant features on a global level (**Syn1**, **Syn2** and **Syn3**) as well as on an instance-wise level (**Syn4**, **Syn5** and **Syn6**) outperforming all other methods in both cases (both global and instance-wise methods). The particularly poor performance of some global feature selection methods in **Syn1**, **Syn2** and **Syn3** (where there is no instance-wise relevance) is due to the non-linearity of the relationship between features and labels, further details can be found in the Appendix.

The results for **Syn4**, **Syn5** and **Syn6** demonstrate that INVASE is capable of detecting a different number of relevant features for each sample when necessary - the performance improvement over L2X is greater in **Syn4** and **Syn5** than **Syn6**. In particular, in **Syn4**, L2X is forced to overselect features when $X_{11} < 0$ and underselect when $X_{11} \geq 0$ thus resulting in higher FDR and lower TPR, respectively. To highlight this, in Table 2 we report the group specific FDR and TPR on **Syn4** and **Syn5** when setting $k = 3, 4, 5$, where Group 1 refers to samples with $X_{11} < 0$ and Group 2 to samples with $X_{11} \geq 0$.

For $k = 3$ in **Syn4**, we see that INVASE and L2X have comparable FDR in Group 1, since the total number of relevant features for each sample is 3 ($X_1, X_2, X_{11}$). However, when we increase $k$, we see that the FDR increases for L2X as it is forced to select more than 3 features, which necessarily means that the FDR must be *at least* 40% even if L2X was finding the relevant features perfectly. On the other hand, for Group 2 we see that the TPR is low for $k = 3$ since necessarily, L2X cannot possibly select all of the 5 relevant features. INVASE, however, is able to select the correct number in both and hence enjoys low FDR and high TPR.

---

[8]Definitions of TPR and FDR can be found in the Appendix.

**Syn5** reinforces the conclusions we drew for L2X in **Syn4**. Interestingly, though, for INVASE, we found that $X_{11}$ was almost never selected for Group 1 in **Syn5**. We believe this is because the lack of overlap between the relevant features for each group means that the predictor network can essentially learn two separate networks - one for each group. This is because it is possible to create two subnetworks with non-overlapping weights that each take as input the features of a given group. $X_{11}$ is therefore unnecessary for prediction. Note, however, that $X_{11}$ is *highly relevant* for the selector network in deciding which features to pass on and so it is not true that $X_{11}$ isn't relevant, but simply that the selector network does not need to "pass on" its relevance to the predictor network. To investigate this further, results for settings where the features overlap between groups (and so it is not possible to disentangle the networks) can be found in the Appendix.

| Datasets | Syn4 | | | | Syn5 | | | |
|---|---|---|---|---|---|---|---|---|
| Group | 1 | | 2 | | 1 | | 2 | |
| Metrics (%) | TPR | FDR | TPR | FDR | TPR | FDR | TPR | FDR |
| **INVASE** | 99.5 | 24.6 | 100.0 | 0.4 | 69.2 | 1.6 | 99.8 | 0.6 |
| L2X ($k = 3$) | 71.1 | 28.9 | 57.2 | 4.6 | 65.5 | 34.5 | 55.4 | 7.7 |
| L2X ($k = 4$) | 81.0 | 39.2 | 74.9 | 6.3 | 76.2 | 42.9 | 72.4 | 9.4 |
| L2X ($k = 5$) | 89.9 | 46.0 | 84.6 | 15.4 | 87.5 | 47.5 | 82.1 | 17.9 |

Table 2: Detailed comparison of INVASE with L2X in **Syn4** and **Syn5**, highlighting the capability of INVASE to select a flexible number of features for each sample. Group 1: $X_{11} < 0$, Group 2: $X_{11} \geq 0$

### 4.1.3 PREDICTION

In this experiment we analyze the effect of using feature selection as a pre-processing step for prediction. We first perform feature selection (either instance-wise or global) and then train a 3-layer fully connected network with Batch Normalization [12] in every layer (to avoid overfitting) to perform predictions on top of the (feature-selected) data. In this setting we compare the two global feature selection methods (LASSO and Tree) and one instance-wise feature selection method (L2X). Furthermore, we also compare with the predictive model without any feature selections (w/o FS) and the predictive model with ground truth globally relevant features[9] (with Global). In particular, this allows us to demonstrate that the improvements in prediction performance are not just because the global feature selection performed implicitly by INVASE is better than the other global feature selection methods but are also due to the fact that we select features on an instance-wise level. Experiments here are conducted on synthetic data with 100 features but the same labelling procedures as above.

As can be seen in Table 3, there is a significant performance improvement when discarding all of the irrelevant features (*with Global*). However, neither of the global feature selection methods (*Tree* and *Lasso*) are capable of achieving this improvement. On the other hand, INVASE is capable of achieving (and beating - in **Syn4** and **Syn6**) this improvement, demonstrating its capability both at selecting features globally better than existing methods but also at improving on global selection with instance-wise selection (where relevant), to provide further improvements. On the other hand, *L2X* performs worse than the global methods in **Syn1-3**, demonstrating an inability to perform even global feature selection in this higher dimensional setting (this is supported by the high dimensional discovery results in the Appendix), and in **Syn4-6** is performing worse than *with Global* (which now is not even optimal).

Furthermore, even though we include Batch Normalization to avoid overfitting, with a small number of samples and high number of dimensions, the 3-layer fully connected network still suffers from overfitting as demonstrated by the significant difference in performance between *w/o FS* and *with Global*. This demonstrates the necessity of feature selection as a pre-processing step. Lastly, in comparison to *with Global*, with INVASE achieves performance gains in **Syn4** and **Syn6**. It quan-

---

[9]For example, in **Syn1** the predictor network in the *with Global* setting is trained on only $X_1$ and $X_2$ and in **Syn4** it would be trained on $X_1, X_2, X_3, X_4, X_5, X_6, X_{11}$.

| Dataset | AUROC | | | | | |
|---------|-------|---|---|---|---|---|
| | w/o FS | with Global | **with INVASE** | with Tree | with L2X | with LASSO |
| **Syn1** | .578±.004 | .686±.005 | .690±.006 | .574±.101 | .498±.005 | .498±.006 |
| **Syn2** | .789±.003 | .873±.003 | .877±.003 | .872±.003 | .823±.029 | .555±.061 |
| **Syn3** | .854±.004 | .900±.003 | .902±.003 | .899±.001 | .862±.009 | .886±.003 |
| **Syn4** | .558±.021 | .774±.006 | .787±.004 | .684±.017 | .678±.024 | .514±.031 |
| **Syn5** | .662±.013 | .784±.005 | .784±.005 | .741±.004 | .709±.008 | .691±.024 |
| **Syn6** | .692±.015 | .858±.004 | .877±.003 | .771±.031 | .827±.017 | .727±.025 |

| Dataset | AUPRC | | | | | |
|---------|-------|---|---|---|---|---|
| | w/o FS | with Global | **with INVASE** | with Tree | with L2X | with LASSO |
| **Syn1** | .567±.007 | .690±.006 | .694±.006 | .577±.102 | .498±.007 | .499±.008 |
| **Syn2** | .799±.005 | .878±.005 | .886±.004 | .878±.004 | .817±.031 | .591±.037 |
| **Syn3** | .861±.003 | .905±.002 | .907±.003 | .904±.002 | .860±.012 | .890±.002 |
| **Syn4** | .572±.019 | .794±.006 | .804±.004 | .681±.031 | .672±.025 | .536±.025 |
| **Syn5** | .665±.019 | .796±.005 | .797±.006 | .765±.003 | .719±.011 | .680±.040 |
| **Syn6** | .709±.018 | .870±.005 | .886±.004 | .779±.027 | .835±.017 | .757±.036 |

Table 3: Prediction performance comparison with and without feature selection methods (L2X, LASSO, Tree, INVASE, and Global). Global is using ground-truth globally relevant features for each dataset

titatively shows that instance-wise feature selection can further improves the predictive model from ground truth global feature selection.

## 4.2 REAL-WORLD DATA EXPERIMENTS

### 4.2.1 DATA DESCRIPTION

In this section we use two real-world datasets to perform a series of further experiments. The first, the Meta-Analysis Global Group in Chronic Heart Failure (MAGGIC) dataset [23], has 40,409 patients each with 31 measured features. The label is all-cause mortality. The second, the Prostate, Lung, Colorectal and Ovarian (PLCO) Cancer Screening Trial in the US and the European Randomized Study of Screening for Prostate Cancer (ERSPC) dataset [8; 26] contains 38,001 each with 106 measured features. The label in this dataset is mortality due to prostate cancer. We refer to this as the PLCO dataset.

The first experiment we carried out was to create semi-synthetic datasets by using the labelling procedures **Syn1-6** from above but with the features now coming from real data (instead of being i.i.d. Gaussian). The results of this experiment can be found in the Appendix.

### 4.2.2 THE DISCOVERED FEATURE IMPORTANCE IN MAGGIC DATASET

In this next experiment, we visualize the ability of INVASE to select features on an individualized level. Fig. 2(left) shows the selection probability (given by INVASE) of each feature for 20 randomly selected patients in the MAGGIC dataset. Fig. 2(right) shows the selection probability of each feature averaged over different binary splits of the data (i.e. when split into Male and Female). In Table 4, we also report the mean and variance of the number of selected features in each subgroup.

As can be seen, INVASE discovers significantly different features for both individuals and for different subgroups of the dataset.

### 4.2.3 RESULTS: PREDICTION USING REAL DATA VARIABLES WITH REAL LABEL

Evaluating the performance of feature selection methods on real data is difficult, since ground truth relevance is often not known. We therefore cannot use TPR and FDR to evaluate the performance on real data. In our final experiment, therefore, we instead focus on prediction performance exactly as in 4.1.3 (except now both the features and label come from real data).

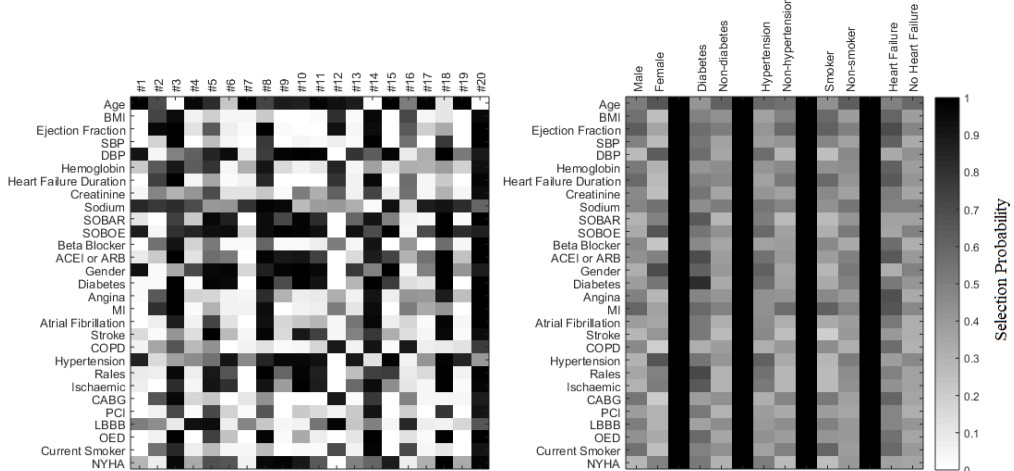

Figure 2: **Left:** The feature importance for each of 20 randomly selected patients in the MAGGIC dataset. **Right:** The average feature importance for different binary splits in the MAGGIC dataset.

| Overall | Male | Diabetes | Hypertension | Smoker | Heart Failure |
|---|---|---|---|---|---|
| | 43.5±10.7 | 53.2±10.8 | 46.6±9.3 | 41.0±12.1 | 51.8±11.1 |
| 42.5±18.4 | Female | Non-diabetes | Non-hypertension | Non-smoker | No Heart Failure |
| | 40.8±15.6 | 39.3±8.0 | 40.0±9.3 | 43.2±7.0 | 39.6±6.9 |

Table 4: Selection probability of overall and patient subgroups by INVASE in MAGGIC dataset. (Mean ± Std)

| **Datasets** | Metrics | AUROC | AUPRC | AUROC | AUPRC |
|---|---|---|---|---|---|
| | Labels | **3 year** | | **5 year** | |
| MAGGIC | INVASE | **.722±.005** | **.655±.010** | **.740±.005** | **.867±.006** |
| | Without INVASE | .720±.006 | .639±.009 | .730±.006 | .855±.004 |
| | Labels | **5 year** | | **10 year** | |
| PLCO | INVASE | **.637±.007** | **.329±.013** | **.673±.007** | **.506±.006** |
| | Without INVASE | .629±.008 | .324±.011 | .657±.006 | .485±.008 |

Table 5: Prediction performance for MAGGIC and PLCO dataset.

As can be seen in Table 5, INVASE consistently improves prediction performance in each of the two settings (different time horizons) in each dataset.

## 5 FUTURE WORK

While this paper has focused on discovering relevant features in the static setting, this could also be extended to apply in the temporal setting. One such avenue of exploration for this would be to replace each of the networks with an RNN. Particular care will need to be taken in defining the problem, though; do we treat each stream as a feature or each time point of each stream? We leave this investigation to future work.

## ACKNOWLEDGEMENT

The authors would like to thank the reviewers for their helpful comments. The research presented in this paper was supported by the Office of Naval Research (ONR) and the NSF (Grant number: ECCS1462245, ECCS1533983, and ECCS1407712).

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

APPENDIX

SUMMARY OF RELATED WORKS

|  | Key ideas | Experiments shown | Global/ Instance-wise | Model agnostic | # of relevant features |
|---|---|---|---|---|---|
| SCFS [11] | Max-dependency min-redundancy criteria with Pearson correlations | Feature selection | Global | Yes | Not needed |
| MIFS [21] | Max-dependency min-redundancy criteria with Mutual Information | Feature selection | Global | Yes | Not needed |
| LASSO [31] | Linear regression with $l_1$-norm penalty | Feature selection Prediction | Global | Yes | Not needed |
| Knock-off [3] | Comparison between knock-off variables and real variables | Feature selection Hypothesis test | Global | Yes | Not needed |
| L2X [4] | Mutual Information maximization with Gumbel-softmax | Interpretation | Instance-wise | Yes | Should be given |
| LIME [24] | Locally linear approximation | Interpretation | Instance-wise | Yes | Should be given |
| Shapley [18] | Shapley value estimation to quantify feature importance | Feature selection | Instance-wise | Yes | Should be given |
| DeepLIFT [27] | Decompose the output of NN on a reference input | Interpretation | Instance-wise | No | Should be given |
| Saliency [29] | Backpropagation from the output of the NN to the input | Interpretation | Instance-wise | No | Should be given |
| Tree SHAP [19] | Shapley value estimation only for tree-ensemble models | Interpretation | Instance-wise | No | Should be given |
| Pixel-wise [1] | Measuring the effects on the output using input perturbation | Interpretation | Instance-wise | No | Should be given |
| **INVASE (Ours)** | Minimize KL divergence using deep NN influenced by actor-critic models | Feature selection Interpretation Prediction | Instance-wise | Yes | Not needed |

Table 6: Summary of the related works. (NN: Neural networks, KL: Kullback-Leibler)

EXTENDING INVASE TO REGRESSION

To extend our model to the setting where $Y$ is continuous (regression problem), we replace the estimated loss with the reconstruction error as follows.

$$\hat{l}(\mathbf{x}, \mathbf{s}) = -||y - f^\phi(\mathbf{x}, \mathbf{s})||_2$$

where $f^\phi : \mathcal{X} \to \mathbb{R}$ is now the (continuous) predictor function trained to minimize the $\ell_2$-norm between its outputs and the real labels. As noted in [9], when the distribution of $Y$ given $\mathbf{X}$ is Gaussian, minimizing the $l_2$-norm is equivalent to minimizing the KL divergence.

DETAILS OF INVASE

In the experiments, the depth of the selector, predictor, and baseline networks is set to 3. The number of hidden nodes in each layer is $d$ and $2d$, respectively. We use either ReLu or SeLu as the activation functions of each layer except for the output layer where we use the sigmoid activation function for the selector network and softmax activation function for the predictor and baseline networks. The number of samples in each mini-batch is 1000 for the selector, predictor, and baseline networks. We use cross-validation to select $\lambda$ among $\{0.1, 0.3, 0.5, 1, 2, 5, 10\}$. We use tensorflow to implement INVASE. The source-code can be found at `https://github.com/iclr2018invase/INVASE/`.

## DETAILS OF BENCHMARKS

We use the following links for the implementations of 7 benchmarks.

- L2X: `https://github.com/Jianbo-Lab/L2X`
- LIME: `https://github.com/marcotcr/lime`
- Shapley: `https://github.com/slundberg/shap`
- Knock-off: `http://web.stanford.edu/group/candes/knockoffs/software/knockoff/`
- Tree: `http://scikit-learn.org/stable/modules/generated/sklearn.ensemble.ExtraTreesClassifier.html`
- LASSO: `http://scikit-learn.org/stable/modules/linear_model.html#lasso`

For L2X, we use the same network settings used in INVASE for fair comparisons. For SCFS, we explicitly implement from the reference ([11]).

## HIGH DIMENSIONAL DISCOVERY

To demonstrate the scalability of our method, we run an experiment in which we increase the total number of features to 100. The features are generated as a 100-dimensional Gaussian with no correlations ($\mathcal{N}(\mathbf{0}, \mathbf{I})$) and the relationships between features and label remains as in Table 1 in the main manucript (i.e. we are adding 89 additional noisy signals that have no effect on the label).

| Dataset | Syn1 | | Syn2 | | Syn3 | | Syn4 | | Syn5 | | Syn6 | |
|---|---|---|---|---|---|---|---|---|---|---|---|---|
| Metrics (%) | TPR | FDR | TPR | FDR | TPR | FDR | TPR | FDR | TPR | FDR | TPR | FDR |
| **INVASE** | **100.0** | **0.0** | **100.0** | **0.0** | **100.0** | **0.0** | **66.3** | **40.5** | **73.2** | **23.7** | **90.5** | **15.4** |
| L2X | 6.1 | 93.9 | 81.4 | 18.6 | 57.7 | 42.3 | 48.5 | 46.4 | 35.4 | 60.8 | 66.3 | 33.7 |
| LIME | 0.0 | 100.0 | **100.0** | **0.0** | 92.7 | 7.3 | 43.8 | 47.4 | 42.3 | 50.1 | 50.1 | 49.9 |
| Shapley | 4.4 | 95.6 | 95.1 | 4.9 | 88.8 | 11.2 | 50.2 | 43.4 | 49.9 | 44.2 | 62.5 | 37.5 |
| Knock off | 0.0 | 64.9 | 3.7 | 71.2 | 74.9 | 24.9 | 28.2 | 59.8 | 33.1 | 59.4 | 46.9 | 53.0 |
| Tree | 49.9 | 50.1 | 100.0 | 0.0 | 100.0 | 0.0 | 40.7 | 49.5 | 56.7 | 37.5 | 58.4 | 41.6 |
| SCFS | 2.5 | 97.5 | 5.3 | 94.7 | 74.9 | 25.1 | 27.0 | 74.6 | 30.6 | 62.1 | 38.3 | 61.7 |
| LASSO | 2.5 | 97.5 | 4.0 | 96.0 | 75.3 | 24.7 | 28.3 | 73.2 | 36.0 | 56.9 | 45.9 | 54.1 |

Table 7: Relevant feature discovery for synthetic datasets with 100 features

As can be seen in Table 7, INVASE also works consistently better than all other benchmarks in all 6 synthetic datasets in this setting. In fact, we see a significant reduction in performance (compared to the 11 feature setting) for L2X in **Syn1**, with the TPR dropping more than 90% leading to an almost complete failure of the method to detect any relevant features. In particular, we see that L2X does not scale as well as INVASE with the dimensionality of the data, which is particularly limiting for a feature selection method.

We also compare the CPU times of the algorithm for training and testing with other instance-wise feature selection benchmarks to show the scalability in terms of computational complexity. As can be seen in Table 8, INVASE is much faster (10 times) than LIME and Shapley methods and comparable with L2X; we see that INVASE takes approximately 50% longer to run than L2X, which can be accounted for by the addition of a 3rd network (the baseline network) in INVASE that is not present in L2X. Note, however, that this baseline network can be trained in parallel with the predictor network and we believe that doing so would lead to both INVASE and L2X having the same run-time.

| Methods | INVASE | L2X | Shapley | LIME |
|---------|--------|-----|---------|------|
| Train | 1327.69s | 939.82s | 12801.21s | - |
| Test | 0.38s | 0.78s | 0.06s | 18931.98s |

Table 8: Comparison of CPU clock time across different instance-wise feature selection methods on average across **Syn1** to **Syn6** with 100 features and 10,000 samples on training/testing, respectively

## HYPER-PARAMETER ANALYSIS

In the following experiment, we provide results for various values of the hyper-parameter, $\lambda$, in the Syn4, Syn5, and Syn6 100-dimensional setting. Table 9 gives the results in terms of TPR and FDR. Note that in the other experiments, we select the hyper-parameter $\lambda$ which maximizes the predictor accuracy in terms of AUROC.

| Datasets | Syn4 | | Syn5 | | Syn6 | |
|----------|------|------|------|------|------|------|
| $\lambda$ / Metris (%) | TPR | FDR | TPR | FDR | TPR | FDR |
| 0.1 | 98.0 | 94.3 | 90.0 | 93.4 | 99.2 | 92.3 |
| 0.3 | 93.7 | 87.9 | 84.2 | 88.9 | 96.9 | 86.7 |
| 0.5 | 99.0 | 43.1 | 88.3 | 50.6 | 99.6 | 31.7 |
| 1 | 66.3 | 40.5 | 73.2 | 23.7 | 90.5 | 15.4 |
| 2 | 0.0 | 0.0 | 25.4 | 4.1 | 67.1 | 3.6 |
| 5 | 0.0 | 0.0 | 7.5 | 2.7 | 7.6 | 2.5 |
| 10 | 0.0 | 0.0 | 0.0 | 0.0 | 0.0 | 0.0 |

Table 9: Relevant feature discovery results for various values of the hyper-parameter $\lambda$ in the Syn4, Syn5, and Syn6 100-dimensional setting.

## ADDITIONAL RESULTS ON COMPLEX SYNTHETIC DATASETS

In the main paper, the relevant subset for a sample in each of our variable synthetic datasets (**Syn4-6**) depended on $X_{11}$ only, which was unused in the rest of the model (i.e. $X_{11}$ determined only the relevant subset, and was otherwise unused as a predictive variable). In this set of experiments, we investigate the effect of having the subset relevance depend on a variable that is also used in the model itself (**Syn4A, Syn5A, Syn6A**). We then investigate the effect of having more than one variable being used to determine subset relevance (**Syn4B, Syn5B, Syn6B, Syn7**). The results for these are reported in Tables 10 and 11, respectively.

The input features are generated from a 100-dimensional Gaussian distribution with no correlations across the features ($\mathbf{X} \sim \mathcal{N}(\mathbf{0}, \mathbf{I})$). $Y$ is generated according to $\mathbb{P}(Y = 1 | \mathbf{X}) = \frac{1}{1 + \text{logit}(\mathbf{X})}$ with the logit value for each synthetic dataset now defined as follows:

- **Syn4A:** If $X_1 < 0$, logit $= \exp(X_1 X_2)$, otherwise, logit $= \exp(\sum_{i=3}^{6} X_i^2 - 4)$.
- **Syn5A:** If $X_1 < 0$, logit $= \exp(X_1 X_2)$, otherwise, logit $= -10 \times \sin 2X_7 + 2|X_8| + X_9 + \exp(-X_{10})$.
- **Syn6A:** If $X_7 < 0$, logit $= \exp(\sum_{i=3}^{6} X_i^2 - 4)$, otherwise, logit $= -10 \times \sin 2X_7 + 2|X_8| + X_9 + \exp(-X_{10})$.

| Dataset | Syn4A | | Syn5A | | Syn6A | |
|---|---|---|---|---|---|---|
| Metrics (%) | TPR | FDR | TPR | FDR | TPR | FDR |
| **INVASE+** | **77.5** | **14.5** | **85.9** | **8.8** | **89.9** | **7.3** |
| L2X | 65.0 | 39.3 | 48.0 | 57.4 | 74.4 | 35.5 |
| LIME | 56.3 | 49.2 | 58.2 | 48.8 | 58.9 | 47.8 |
| Shapley | 71.8 | 39.8 | 71.0 | 41.3 | 68.9 | 38.2 |
| Knock off | 59.8 | 62.6 | 55.0 | 49.9 | 65.0 | 40.0 |
| Tree | 61.3 | 46.9 | 75.6 | 39.4 | 66.9 | 40.0 |
| SCFS | 52.8 | 66.9 | 55.3 | 50.6 | 50.4 | 51.8 |
| LASSO | 61.0 | 61.2 | 55.0 | 50.0 | 53.9 | 48.8 |

Table 10: Relevant feature discovery results for complex synthetic datasets (**Syn4A, 5A, 6A**) with 100 features

- **Syn4B:** If $X_1 X_3 < 0$, logit $= \exp(X_1 X_2)$, otherwise, logit $= \exp(\sum_{i=3}^{6} X_i^2 - 4)$.
- **Syn5B:** If $X_1 X_7 < 0$, logit $= \exp(X_1 X_2)$, otherwise, logit $= -10 \times \sin 2X_7 + 2|X_8| + X_9 + \exp(-X_{10})$.
- **Syn6B:** If $X_3 X_7 < 0$, logit $= \exp(\sum_{i=3}^{6} X_i^2 - 4)$, otherwise, logit $= -10 \times \sin 2X_7 + 2|X_8| + X_9 + \exp(-X_{10})$.
- **Syn7:**
  - If $X_1 < 0, X_2 < 0$, logit $= \exp(X_1 X_2)$
  - If $X_1 < 0, X_2 \geq 0$, logit $= \exp(\sum_{i=3}^{6} X_i^2 - 4)$.
  - If $X_1 \geq 0, X_2 < 0$, logit $= -10 \times \sin 2X_7 + 2|X_8| + X_9 + \exp(-X_{10})$.
  - If $X_1 \geq 0, X_2 \geq 0$, logit $= 0.5 \times \exp(X_1 X_2) + 0.5 \times \exp(\sum_{i=3}^{4} X_i^2 - 2)$.

## RESULTS ON SEMI-SYNTHETIC DATASETS

In this experiment, we use real features (which have correlation across features) but generate the labels as in the synthetic experiments from the main paper, using **Syn1-Syn6**. This allows us to know the ground truth relevance of the features, and calculate TPR and FDR, while using unknown and correlated feature distributions (instead of the unrealistic setting of i.i.d. Gaussian used in the fully synthetic experiment). The results for the MAGGIC and PLCO datasets are given below.

| Dataset | Syn4B | | Syn5B | | Syn6B | | Syn7 | |
|---|---|---|---|---|---|---|---|---|
| Metrics (%) | TPR | FDR | TPR | FDR | TPR | FDR | TPR | FDR |
| **INVASE+** | **65.5** | **30.1** | **85.0** | **15.0** | **86.5** | **27.8** | **86.8** | **32.4** |
| L2X | 43.2 | 53.4 | 50.3 | 50.3 | 44.6 | 55.4 | 35.3 | 70.9 |
| LIME | 56.8 | 37.2 | 71.9 | 27.2 | 69.8 | 30.2 | 56.4 | 51.0 |
| Shapley | 51.4 | 43.5 | 77.2 | 24.1 | 69.3 | 30.7 | 61.8 | 45.6 |
| Knock off | 5.3 | 87.4 | 73.3 | 25.2 | 59.9 | 40.1 | 54.1 | 60.0 |
| Tree | 56.7 | 37.4 | 73.9 | 25.0 | 70.1 | 29.9 | 71.6 | 40.3 |
| SCFS | 3.7 | 96.2 | 72.3 | 26.3 | 61.1 | 38.9 | 22.9 | 77.5 |
| LASSO | 4.2 | 95.6 | 73.3 | 25.0 | 60.1 | 39.9 | 24.9 | 75.8 |

Table 11: Relevant feature discovery results for complex synthetic datasets (**Syn4B, 5B, 6B, 7**) with 100 features

| Dataset | Syn1 | | Syn2 | | Syn3 | | Syn4 | | Syn5 | | Syn6 | |
|---|---|---|---|---|---|---|---|---|---|---|---|---|
| Metrics (%) | TPR | FDR | TPR | FDR | TPR | FDR | TPR | FDR | TPR | FDR | TPR | FDR |
| **INVASE** | **100.0** | **0.0** | **100.0** | **0.0** | **100.0** | **0.0** | **85.9** | **0.0** | **72.9** | **0.1** | **81.0** | **13.2** |
| L2X | 68.8 | 31.2 | 99.9 | 0.1 | 83.0 | 17.0 | 60.0 | 31.3 | 68.3 | 22.3 | 73.5 | 26.5 |
| LIME | 46.9 | 53.1 | 99.9 | 0.1 | 87.2 | 12.8 | 63.6 | 24.4 | 50.2 | 37.6 | 68.7 | 31.3 |
| Shapley | 73.9 | 26.1 | 94.5 | 5.5 | 81.0 | 19.0 | 65.3 | 23.9 | 61.2 | 29.0 | 69.9 | 30.1 |
| Knock off | 27.5 | 65.0 | 77.5 | 22.5 | **100.0** | **0.0** | 57.0 | 34.4 | 56.1 | 29.8 | 58.0 | 42.0 |
| Tree | **100.0** | **0.0** | **100.0** | **0.0** | **100.0** | **0.0** | 56.3 | 29.7 | 51.6 | 40.2 | 46.7 | 53.3 |
| SCFS | 30.0 | 70.0 | 53.0 | 47.0 | **100.0** | **0.0** | 52.0 | 39.9 | 54.0 | 32.4 | 64.5 | 35.5 |
| LASSO | 25.0 | 75.0 | 75.0 | 25.0 | **100.0** | **0.0** | 60.7 | 33.1 | 56.1 | 29.8 | 58.2 | 41.8 |

Table 12: Relevant feature discovery for real datasets with synthetic labels using MAGGIC dataset

| Dataset | Syn1 | | Syn2 | | Syn3 | | Syn4 | | Syn5 | | Syn6 | |
|---|---|---|---|---|---|---|---|---|---|---|---|---|
| Metrics (%) | TPR | FDR | TPR | FDR | TPR | FDR | TPR | FDR | TPR | FDR | TPR | FDR |
| **INVASE** | **35.9** | **0.0** | **100.0** | **0.0** | 84.0 | 7.0 | **59.2** | **38.6** | **64.6** | **31.7** | **70.0** | **29.9** |
| L2X | 0.0 | 100.0 | 62.2 | 37.8 | 43.6 | 56.4 | 41.9 | 55.4 | 21.5 | 76.7 | 66.9 | 33.1 |
| LIME | 1.0 | 99.0 | 70.3 | 29.7 | 74.9 | 25.1 | 43.5 | 55.9 | 26.8 | 68.9 | 56.8 | 43.2 |
| Shapley | 5.4 | 94.6 | 68.5 | 31.5 | 67.9 | 32.1 | 32.7 | 69.4 | 39.6 | 58.6 | 48.5 | 51.5 |
| Knock off | 15.0 | 50.0 | 85.0 | 15.0 | **100.0** | **0.0** | 46.1 | 52.1 | 34.5 | 58.3 | 60.0 | 40.0 |
| Tree | 0.0 | 100.0 | 71.0 | 29.0 | 75.0 | 25.0 | 34.5 | 66.3 | 43.8 | 54.7 | 36.9 | 63.1 |
| SCFS | 10.0 | 90.0 | 61.0 | 39.0 | 93.8 | 6.2 | 43.2 | 55.7 | 31.0 | 63.6 | 55.5 | 44.5 |
| LASSO | 0.0 | 100.0 | 72.5 | 27.5 | **100.0** | **0.0** | 39.2 | 60.8 | 33.2 | 68.2 | 45.0 | 55.0 |

Table 13: Relevant feature discovery for real datasets with synthetic labels using PLCO dataset

As demonstrated in Tables 12 and 13, INVASE outperforms all other methods across all 6 of the synthetic-label settings using real features. This also demonstrates the capability of INVASE in settings where there are unknown correlation structures in the features.

## 5.1 PREDICTIVE PERFORMANCE COMPARISON ON REAL-WORLD DATASETS

In this experiment, we evaluate the predictive performance gains of using each feature selection method as a pre-processing step on the two real datasets, MAGGIC and PLCO (as was done for synthetic data in Section 4.1.3). For each method, we first perform feature selection and then train a predictive model on top of the feature-selected data, where the model has the same architecture as the INVASE predictor network (to create a fair comparison of methods). As can be seen in Table 14, INVASE significantly outperform the other approaches.

| Datasets | MAGGIC | | | | PLCO | | | |
|---|---|---|---|---|---|---|---|---|
| Labels | 3-year | | 5-year | | 5-year | | 10-year | |
| Metrics | AUROC | AUPRC | AUROC | AUPRC | AUROC | AUPRC | AUROC | AUPRC |
| INVASE | 0.722 | 0.655 | 0.740 | 0.867 | 0.637 | 0.329 | 0.673 | 0.506 |
| L2X | 0.609 | 0.529 | 0.607 | 0.794 | 0.558 | 0.170 | 0.583 | 0.365 |
| LIME | 0.637 | 0.5596 | 0.634 | 0.808 | 0.597 | 0.183 | 0.601 | 0.374 |
| Shapley | 0.641 | 0.557 | 0.617 | 0.797 | 0.614 | 0.194 | 0.615 | 0.381 |
| Knockoff | 0.686 | 0.614 | 0.711 | 0.853 | 0.619 | 0.230 | 0.658 | 0.475 |
| Tree | 0.678 | 0.604 | 0.708 | 0.850 | 0.632 | 0.269 | 0.655 | 0.469 |
| SCFS | 0.683 | 0.623 | 0.723 | 0.857 | 0.632 | 0.231 | 0.632 | 0.444 |
| LASSO | 0.692 | 0.615 | 0.709 | 0.847 | 0.623 | 0.218 | 0.656 | 0.467 |

Table 14: Predictive Performance Comparison on two real-world datasets (MAGGIC and PLCO) in terms of AUROC and AUPRC

CORRELATIONS BETWEEN FEATURES AND LABELS IN THE SYNTHETIC AND
SEMI-SYNTHETIC EXPERIMENTS

| Variables | Syn1 | Syn2 | Syn3 | Syn4 | Syn5 | Syn6 |
|---|---|---|---|---|---|---|
| $X_1$ | **0.003** | 0.008 | 0.006 | **0.009** | **0.007** | 0.006 |
| $X_2$ | **0.001** | 0.005 | 0.006 | **0.005** | **0.015** | 0.005 |
| $X_3$ | 0.006 | **0.011** | 0.001 | **0.017** | 0.016 | **0.010** |
| $X_4$ | 0.006 | **0.003** | 0.003 | **0.002** | 0.000 | **0.002** |
| $X_5$ | 0.003 | **0.015** | 0.022 | **0.004** | 0.017 | **0.028** |
| $X_6$ | 0.003 | **0.004** | 0.005 | **0.002** | 0.004 | **0.005** |
| $X_7$ | 0.013 | 0.009 | **0.481** | 0.002 | **0.242** | **0.235** |
| $X_8$ | 0.010 | 0.008 | **0.012** | 0.003 | **0.010** | 0.022 |
| $X_9$ | 0.001 | 0.003 | **0.239** | 0.002 | **0.115** | **0.121** |
| $X_{10}$ | 0.002 | 0.003 | **0.308** | 0.003 | **0.149** | **0.144** |
| $X_{11}$ | 0.014 | 0.012 | 0.004 | **0.028** | **0.018** | **0.002** |

Table 15: Correlation between features and labels in Synthetic datasets with 100 features. Ground truth (in the global sense) relevant features are given in bold. Features with correlation $> 0.05$ are highlighted in red.

As can be seen in Table 15, among 33 relevant features, only 9 features have more than 0.05 (linear) correlation with the label. In particular, using a linear model, it is very hard to discover the relevant features. However, Knock-off (based on LASSO and linear correlations), LASSO, and SCFS are linear models, resulting in a poor performance in our experiments. The above table results are directly reflected in the results given in the main manuscript.

| Variables | Syn1 | Syn2 | Syn3 | Syn4 | Syn5 | Syn6 |
|---|---|---|---|---|---|---|
| $X_1$ | **0.028** | 0.030 | 0.070 | **0.011** | **0.044** | 0.026 |
| $X_2$ | **0.002** | 0.011 | 0.009 | **0.001** | **0.001** | 0.012 |
| $X_3$ | 0.018 | **0.079** | 0.008 | **0.038** | 0.006 | **0.046** |
| $X_4$ | 0.005 | **0.113** | 0.006 | **0.056** | 0.001 | **0.055** |
| $X_5$ | 0.006 | **0.034** | 0.032 | **0.013** | 0.016 | **0.036** |
| $X_6$ | 0.019 | **0.114** | 0.027 | **0.099** | 0.018 | **0.004** |
| $X_7$ | 0.005 | 0.010 | **0.367** | 0.000 | **0.262** | **0.272** |
| $X_8$ | 0.020 | 0.030 | **0.112** | 0.023 | **0.075** | **0.082** |
| $X_9$ | 0.025 | 0.022 | **0.299** | 0.006 | **0.216** | **0.200** |
| $X_{10}$ | 0.07 | 0.043 | **0.328** | 0.027 | **0.222** | **0.206** |
| $X_{11}$ | 0.009 | 0.006 | 0.034 | **0.046** | **0.018** | 0.058 |

Table 16: Correlation between features and labels in MAGGIC datasets. Ground truth relevant features are described in bold. Features with correlation $> 0.05$ are described in red

We do the same analysis for the MAGGIC dataset; results are given in Table 16. We see that here the linear correlation with the label is stronger and this is reflected in Tables 12 and 13, where all of the linear models performed better than in the fully-synthetic settings. However, we note that although they had a better performance, in most cases it was still not comparable with INVASE.

DEFINITION OF TPR AND FDR

| | | True Condition | |
|---|---|---|---|
| | | Positive | Negative |
| Predicted Condition | Positive | **True Positive** | **False Positive** |
| | Negative | **False Negative** | **True Negative** |

$$\textbf{True Positive Rate (TPR)}= \frac{\text{True Positive}}{\text{True Positive+False Negative}}$$

$$\textbf{False Discovery Rate (FDR)}= \frac{\text{False\ Positive}}{\text{True Positive+False Positive}}$$

Figure 3: The definitions of True Positive Rate (TPR) and False Discovery Rate (FDR)

## COMPUTER VISION

Another natural application of INVASE is in computer vision. To briefly demonstrate the applicability and capability of INVASE to computer vision, we conduct two experiments using the Kaggle Dogs vs. Cats dataset (`https://www.kaggle.com/c/dogs-vs-cats`) [6] and the Oxford Pet dataset (`http://www.robots.ox.ac.uk/~vgg/data/pets/`) [20]. The goal is to select a set 16 x 16 patches of each image that maximize the predictive capability of a model. In order to apply INVASE to this problem, we simply treat each 16 x 16 patch as a feature.

We use the U-Net [25] architecture for the selector network and the VGG network [28] architecture for the predictor and baseline networks. Below we give qualitative results of INVASE applied to these datasets, where we see that INVASE successfully identifies patches of each image in which the animal's face is visible.

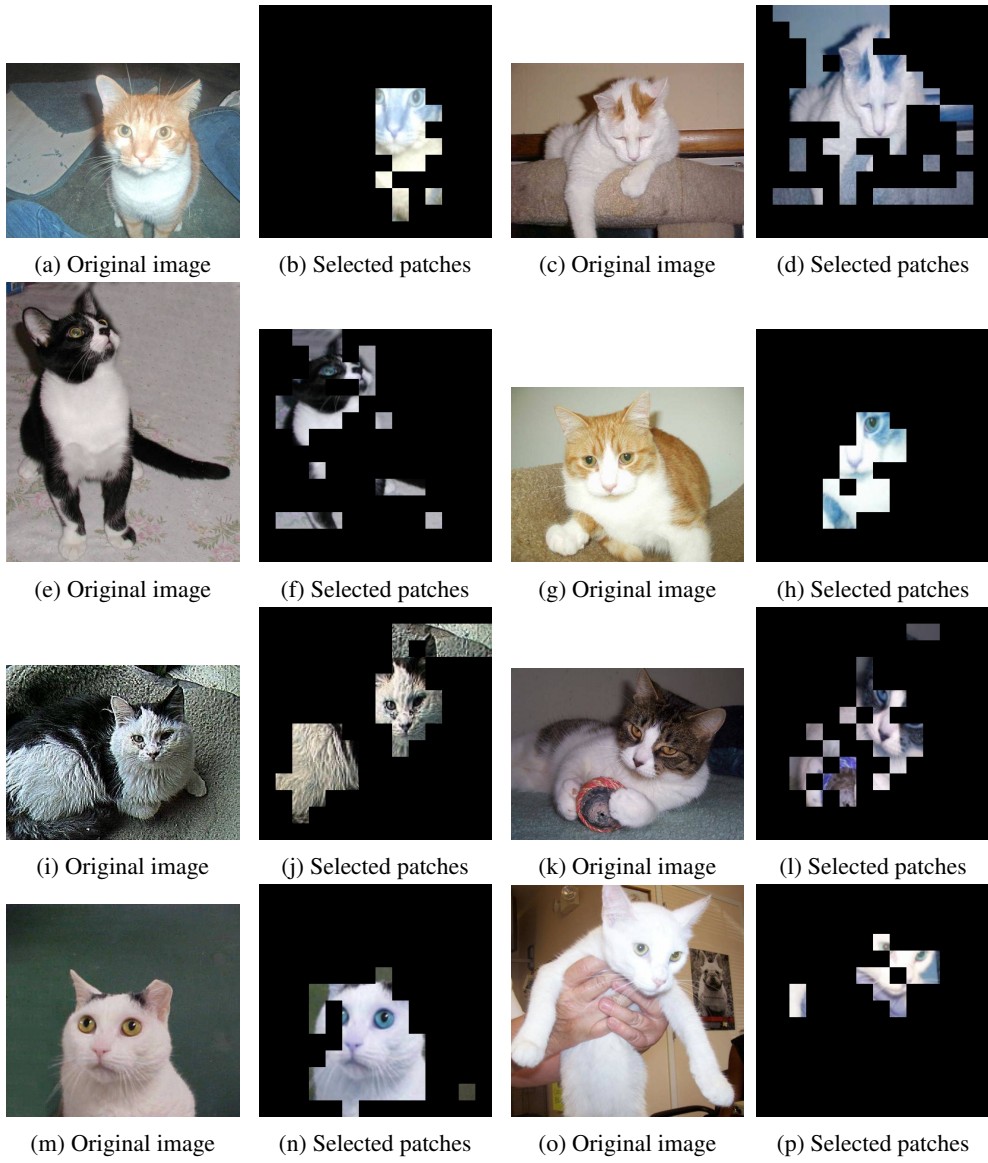

(a) Original image    (b) Selected patches    (c) Original image    (d) Selected patches

(e) Original image    (f) Selected patches    (g) Original image    (h) Selected patches

(i) Original image    (j) Selected patches    (k) Original image    (l) Selected patches

(m) Original image    (n) Selected patches    (o) Original image    (p) Selected patches

Figure 4: Selected 16 x 16 patches by INVASE on Kaggle Dogs vs. Cats dataset - Cats

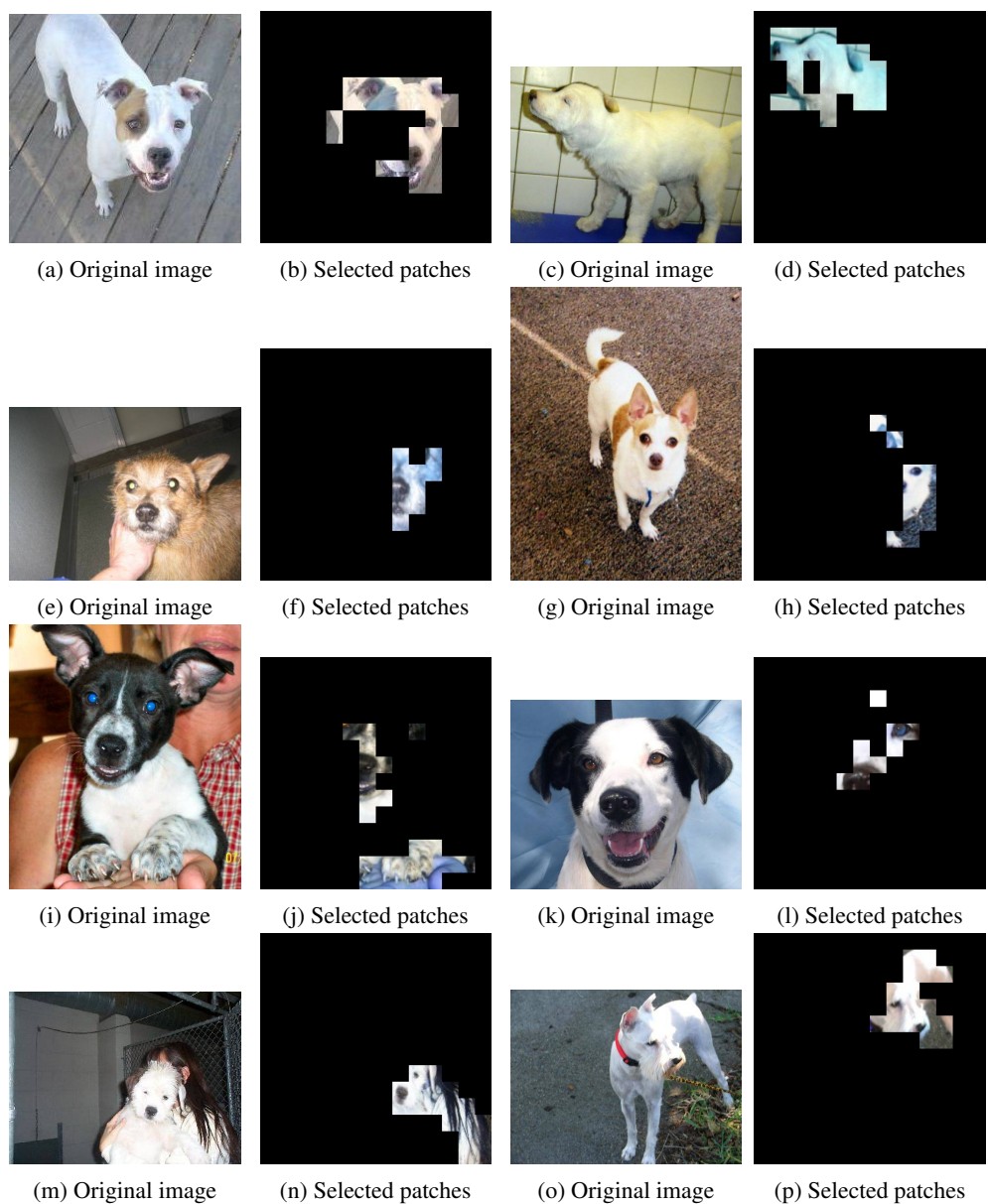

Figure 5: Selected 16 x 16 patches by INVASE on Kaggle Dogs vs. Cats dataset - Dogs

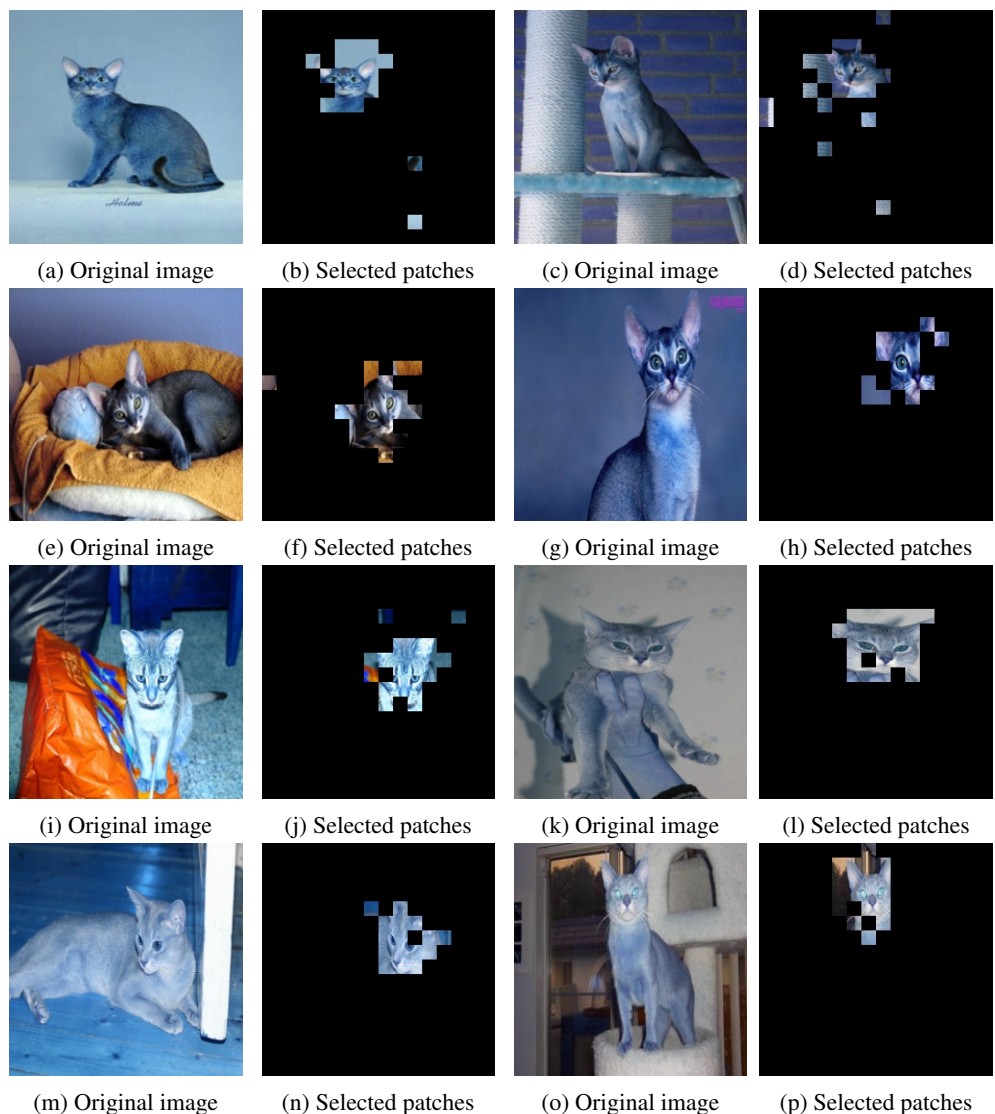

Figure 6: Selected 16 x 16 patches by INVASE on Oxford Pet dataset - Cats

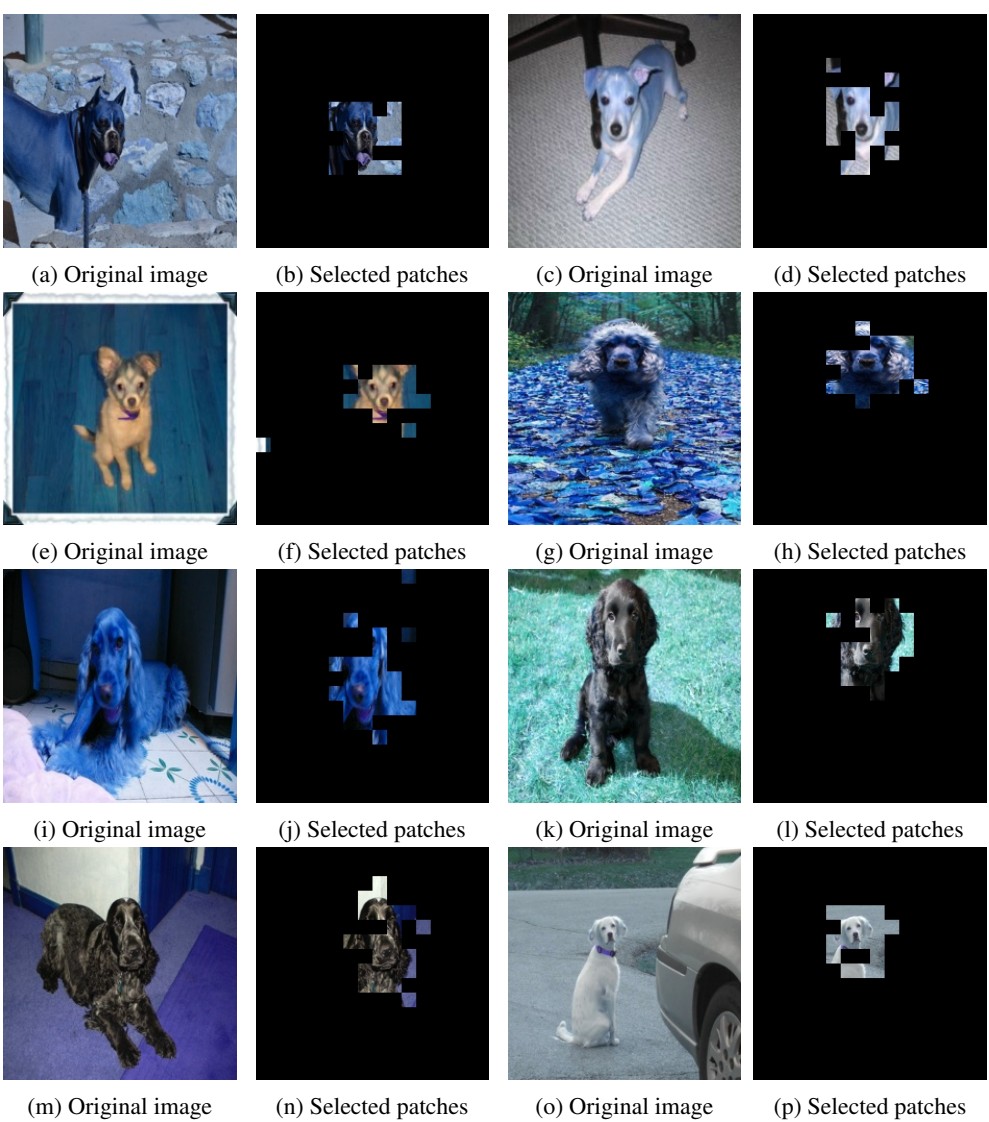

(a) Original image    (b) Selected patches    (c) Original image    (d) Selected patches

(e) Original image    (f) Selected patches    (g) Original image    (h) Selected patches

(i) Original image    (j) Selected patches    (k) Original image    (l) Selected patches

(m) Original image    (n) Selected patches    (o) Original image    (p) Selected patches

Figure 7: Selected 16 x 16 patches by INVASE on Oxford Pet dataset - Dogs

