# OpenReview forum: "INVASE: Instance-wise Variable Selection using Neural Networks"
_ICLR.cc/2019/Conference_

### Official Review · AnonReviewer2 · 2018-10-24
**Simple idea, but good results**

**Rating:** 6
**Confidence:** 3

**Review:**

In the paper, the authors proposed a new algorithm for instance-wise feature selection. In the proposed algorithm, we prepare three DNNs, which are predictor network, baseline network, and selector network. The predictor network and the baseline networks are trained so that it fits the data well, where the predictor network uses only selected features sampled from the selector network. The selector network is trained to minimize the KL-divergence between the predictor network and the baseline network. In this way, one can train the selector network that select different feature sets for each of given instances.

I think the idea is quite simple: the use of three DNNs and the proposed loss functions seem to be reasonable. The experimental results also look promising.

I have a concern on the scheduling of training. Too fast training of the predictor network can lead to the subotpimal selection network. I have checked the implementations in github, and found that all the networks used Adam with the same learning rates. Is there any issue of training instability? And, if so, how we can confirm that good selector network has trained?

My another concern is on the implementations in github. The repository originally had INVASE.py. In the middle of the reviewing period, I found that INVASE+.py has added. I am not sure which implementations is used for this manuscript. It seems that INVASE.py contains only two networks, while INVASE+.py contains three networks. I therefore think the latter is the implementation used for this manuscript. If this is the case, what INVASE.py is for?
I am also not sure if it is appropriate to "communicate" through external repositories during the reviewing period.

---

> ### Author Response · Authors · 2018-11-15
> **RE: Simple idea, but good results**
>
> Thank you for the insightful comments.
>
> A1: It is not true that fast training of the predictor network can lead to a suboptimal selector network. Even when the predictor network is fully trained after each selector network update, the selector network can converge optimally. However, because the input distribution of the predictor network changes with each update of the selector network, the predictor network will have to update after each selector update. It is therefore not possible for the predictor network to converge until after the selector network has converged. Therefore, there are no stability issues caused by using the same learning rates for each network.
>
> A2: INVASE+.py is the code corresponding to the implementation found in this paper. INVASE.py corresponds to the same implementation but without the baseline (i.e. just the selector and predictor networks). In practice we found both to perform similarly, but the derivation of INVASE+ is a little more natural, and as such we used it for the paper.
>
> We have since changed the names in the repository to INVASE and INVASE- (so that now INVASE is indeed the implemented method and INVASE- is the method without the baseline). We hope this alleviates the confusion.

---

### Official Review · AnonReviewer1 · 2018-11-02

**Rating:** 6
**Confidence:** 4

**Review:**

This paper proposes a new instance-wise feature selection method, INVASE. It is closely related to the prior work L2X (Learning to Explain). There are three differences compared to L2X. The most important difference is about how to backpropagate through subset sampling to select features.  L2X use the Gumbel-softmax trick and this paper uses actor-critic models.

The paper is written well. It is easy to follow the paper. The contribution of this paper is that it provides a new way,  compared to L2X, to backpropagate through subset sampling in order to select features. The authors compare INVASE with L2X and several other approaches on synthetic data and show outperforming results. In the real-world experiments, the authors do not compare INVASE with other approaches.

Regarding experiments, instance-wise feature selection is often applied on computer vision or natural language process applications, where global feature selection is not enough. This paper lacks experiments on CV or NLP applications. For the MAGGIC dataset, I expect to see subgroup patterns. The patterns that authors show in Figure 2 are very different for all randomly selected 20 patients. The authors do not explain why it is preferred to see very different feature patterns for all patients instead of subgroup patterns.

I have questions about other two differences from L2X, pointed by the authors. First, the selector function outputs a probability for selecting each feature \hat{S}^\theta(x). In the paper of L2X, it also produces a weight vector w_\theta(x) as described in section 3.4. I think the \hat{S}^\theta(x) has similar meaning as w_\theta(x) in L2X. In the synthetic data experiment, the authors fix the number of selected features for L2X so that it forces to overselect or underselect features in the example of Syn4. Did the author try to relax this constraint for L2X and use w_\theta(x) in L2X to select features as using \hat{S}^\theta(x) in INVASE?

Second, I agree with the authors that L2X is inspired by maximizing mutual information between Y and X_S and INVASE is inspired by minimizing KL divergence between Y|X and Y|X_S. Both intuitions lead to similar objective functions that INVASE has an extra term \log p(y|x) and \lambda ||S(x)||. INVASE is able to add a l_0 penalty on S(x) since it uses the actor-critic models. For the \log p(y|x) term, as the author mentioned, it helps to reduce the variance in actor-critic models. This \log p(y|x) term is a constant in the optimization of S(x). In Algorithm 1, 12, the updates of \gamma does not depend on other parameters related to the predictor network and selector network. Could the authors first train a baseline network and use it as a fixed function in Algorithm 1? I don't understand the meaning of updates for \gamma iteratively with other parameters since it does not depend on the learning of other parameters. Does this constant term \log p(y|x) have other benefits besides reducing variance in actor-critic models?

I have another minor question about scaling. How does the scaling of X affect the feature importance learned by INVASE?

Note: I have another concern about the experiments. Previous instance-wise variable selection methods are often tested on CV or NLP applications, could the authors present those experiments as previous works?

---

> ### Author Response · Authors · 2018-11-15
> **RE2: Review**
>
>
> A4: The main problem in doing this for L2X is in the training stage (not the testing stage). As can be seen in the equations to compute V values (page 4 end of the left column of L2X paper), they must provide some k to train with which is, in general, unknown in real-world datasets (because we don’t know how many features are relevant in the real-world datasets). The weights w(X) are optimized according to a specific feature selection strategy during training, using them in a different strategy during testing would not make sense, as they are no longer optimized for this strategy. While intuitively possible, consider that, due to the way they’ve been trained, the weights w(X) are expected to “spit out” k features. Because of this, it might be that the weights for the unselected k features are essentially random (but lower than the selected k features). We have no reason to believe that the weights beyond the selected k features would be meaningful (since during training the method only ever selected precisely k features).
>
> We have, however, conducted an experiment in the Syn4 and Syn5 settings with 100 featurs in which we directly use w(X) and threshold it to select features. As can be seen below, the results are significantly worse than for INVASE and the large increase in FDR is indeed consistent with the fact that the weights beyond the top k are not well-disciplined. We will clarify this in the revised manuscript. Note that the published code of the L2X paper is also forced to select k features in both the training and testing stages.
> ----------------------------------------------------------------------------------------
>          Datasets             |               Syn4            |             Syn5              |
>          Thresholds         |      TPR     |    FDR    |      TPR    |     FDR     |
> ----------------------------------------------------------------------------------------
>       L2X      |      0.1      |      87.4     |     93.5   |     79.5     |     95.3    |
>       L2X      |      0.3      |      69.9     |     83.8   |     77.2     |     77.1    |
>       L2X      |      0.5      |      69.8     |     64.1   |     66.4     |     84.6    |
>       L2X      |      0.7      |      59.1     |     61.2   |     54.4     |     65.7    |
>       L2X      |      0.9      |      52.7     |     44.8   |     51.2     |     50.5    |
>              INVASE           |       66.3    |      40.5   |     73.2     |     23.7    |
> ----------------------------------------------------------------------------------------
>
> A5: The baseline network does not have to be trained iteratively with the other networks, however in actor-critic models it typically is. This is because the baseline is used in some sense to “normalize” the predictor network. For this reason, it is therefore good to have the baseline and predictor at a similar “level of convergence”. However, the performance differences are marginal between the two methods, and so we found that it was not important which training method we used.
>
> A6: The scaling of X is not important. At no point do we multiply the feature vector (X) by the “importance weights”. The weights are used to obtain a binary mask vector which is then multiplied (element-wise) with the feature vector. As such, the unselected features end up being 0 and the selected features retain their original value.

---

> > ### Comment · AnonReviewer1 · 2018-11-16
> > **The authors addressed my concerns in the technical aspect**
> >
> > First, I want to thank the authors for providing detailed replies to my questions. I have detailed comments for how I think of this paper in my original review. The authors have done a good work to address my concerns. I write my current opinions in short to support my updated score. I see the value of this work is proposing a new instance-wise feature selection method, INVASE, which has a tight relation with L2X. The most important value of INVASE compared to L2X is that one does not need to choose k, the number of relevant features, in advance. The authors have demonstrated that INVASE outperforms L2X and other methods for their synthetic data. They use MAGGIC medical dataset to show that INVASE is doing instance-wise feature selection qualitatively. The authors mention in the feedback that they will run experiments on Kaggle Dog vs Cat dataset and provide those results in the Appendix. I think the work has the value that it does not need to choose k in advance compared to L2X and the authors have detailed synthetic data results, so I tend to accept this work and updated my score. I do not see that instance-wise feature selection is useful for medical dataset MAGGIC in practice and I think that instance-wise feature selection is useful for CV and NLP applications. The lack of CV and NLP applications is a weak point. It is nice to include Kaggle Dog vs Cat dataset in the Appendix, the authors can consider applications in L2X and other previous works as well.
> >
> > One can check my original questions and authors' feedback for experiments on synthetic dataset and MAGGIC dataset. I write a summary about my technical questions and authors' feedback. I think that they addressed my concerns. First question is whether one can use w_\theta(x) in L2X to select features. As can be seen in authors' feedback, w_\theta(x) in L2X has to depend on k and it does not work well. Second question is the meaning of the baseline network. Though it is a constant term in optimization, it is used to reduce variance in actor-critic models. Intuitively, the update of the baseline network can help to get predict network and baseline network at a similar "level of convergence" in each state. As the authors mentioned, if one trains an optimal baseline network and use it, the performance difference is marginal. The authors have addressed my technical concerns. I am satisfied with the feedback.

---

> ### Author Response · Authors · 2018-11-15
> **RE1: Review**
>
> Thank you for the insightful comments.
>
> A1: We performed extensive experiments in the synthetic setting on all methods (we both reproduced and extended the settings from L2X). In addition to this, results for semi-synthetic data (where the underlying features are from real data but the label is generated synthetically) can be found in the Appendix on page 16. It is necessary to perform experiments on synthetic data if we wish to be able to compare the TPR and FDR of the different methods since we require knowledge of the ground truth relevant features.
>
> For the real-world results, our focus was on qualitative results (believing we had already demonstrated the methods efficacy in the synthetic - and in the appendix the semi-synthetic - settings). We will move the semi-synthetic results to the main body of the paper to make clear that we have demonstrated the performance in this setting.
>
> For the real-data experiment in which we report prediction performance, we have extended our results to include the other approaches. We use the same predictive model as the INVASE predictor network (to allow a fair comparison) but use only the selected features of each approach. As can be seen in the below table (for the PLCO dataset), INVASE does significantly outperform the other approaches. Detailed results will be added to the revised manuscript.
>
> ----------------------------------------------------------------------------------------------------
>          Labels           |                  5-year                 |                  10-year                |
>          Metrics         |      AUROC     |    AUPRC    |      AUROC     |     AUPRC    |
> ----------------------------------------------------------------------------------------------------
>         INVASE           |     0.637         |     0.329      |       0.673        |       0.506    |
>            L2X               |     0.558         |     0.170      |       0.583        |       0.365    |
>           LIME             |     0.597         |     0.183      |       0.601        |       0.374    |
>         Shapley          |     0.614         |     0.194      |       0.615        |       0.381    |
>         Knockoff        |     0.619         |     0.230      |       0.658        |       0.475    |
>            Tree             |     0.632         |     0.269      |       0.655        |       0.469    |
>           SCFS             |     0.632         |     0.231      |       0.632        |       0.444    |
>           LASSO          |     0.623         |     0.218      |       0.656        |       0.467    |
> ----------------------------------------------------------------------------------------------------
>
> A2: Our method can definitely be applied to CV or NLP, though in the paper we focus on what we believe to be an equally important application where global feature selection is not enough, i.e. medicine.
> We will provide qualitative results in the Appendix of the revised manuscript using the Kaggle Dog vs Cat dataset (https://www.kaggle.com/c/dogs-vs-cats).
>
> A3: The results shown in figure 2 for the MAGGIC dataset are entirely qualitative. We are not suggesting that the patterns shown are preferred (or expected) but rather showing that when we use INVASE to discover features for MAGGIC, we find that the patterns are different (though, if you look at, for example, patients 9, 10 and 11 we see a similar pattern for all 3). To us, this simply reinforces the fact that instance-wise feature selection is necessary - if MAGGIC did indeed only contain subgroup patterns then we would expect INVASE to pick these out (as it does in the synthetic and semi-synthetic experiments where, for example in Syn4, Syn5 and Syn6, there are two distinct subgroups).

---

### Official Review · AnonReviewer3 · 2018-11-04
**Instance-wise feature selection**

**Rating:** 6
**Confidence:** 3

**Review:**

This paper proposes an instance-wise feature selection method, which chooses relevant features for each individual sample. The basic idea is to minimize the KL divergence between the distribution p(Y|X) and p(Y|X^{(s)}). The authors consider the classification problem and construct three frameworks: 1) a selector network to calculate the selection probability of each feature; 2) a baseline network for classification on all features; 3) a predictor network for classification on selected features. The goal is to minimize the difference between the baseline loss and predictor loss.

The motivation of the paper is clear and the presentation is easy to follow. However, I have some questions on the model and experiments:

1. How is Eq. (5) formulated? As the selector network does not impact the baseline network, an intuition regarding Eq. (5) is to maximize the predictor loss, which seems not reasonable. It seems more appropriate to use an absolute value of the difference in Eq. (5). Some explanation for the formulation of Eq. (5) would be helpful.

2. The model introduces an extra hyper-parameter, $\lambda$, to adjust the sparsity of selected features. I was curious how sensitive is the performance w.r.t. this hyper-parameter. How is $\lambda$ determined in the experiments?

3. After the selector network is constructed, how are the features selected on testing data? Is the selection conducted by sampling from the Bernoulli distribution as in training or by directly cutting off the features with lower probabilities?

---

> ### Author Response · Authors · 2018-11-15
> **RE: Instance-wise feature selection**
>
> Thank you for the insightful comments.
>
> A1: Equation (5) is the difference between the cross-entropies of the predictor and baseline networks. The first term (-sum_y log f_i^\phi (x^(s), s)) is the cross-entropy of the predictor network and the second term (-sum y log f_i^\gamma (x)) is the cross-entropy of the baseline network. The loss in equation (5) is defined as the “first term – second term”. The selector network is trained to minimize this, not maximize it. Note that the baseline network is introduced to reduce the variance of this quantity, and not as a term that the selector network can change (this is a standard technique used in the actor-critic literature).
>
> Also note that if the baseline network term (the second term) in equation (5) is removed, then we simply end up with the predictor loss defined in the “Predictor Network” section (l_1).
>
> If instead we were to use absolute value, then when the baseline network loss is larger than the predictor network loss, the method would actually be trying to maximise the predictor network loss (which we do not want).
>
> It is important to note that we are not trying to minimize the difference between the predictor and baseline losses - we are using the baseline to reduce the variance of the overall loss and we are simply trying to minimize the predictor loss.
>
> A2: As can be seen in page 13 (subsection “Details of INVASE”), we explain that “We use cross-validation to select lambda among {0.1,0.3,0.5,1,2,5,10}”. We select the lambda which maximizes the predictor accuracy in terms of AUROC. We will clarify this in the revised manuscript. Below, we give the results for various values of lambda in the Syn4, Syn5, and Syn6 settings. More detailed results will be added to the revised manuscript.
> --------------------------------------------------------------------------------------------------------------------------------------------------
>                Datasets             |                   Syn4                   |                   Syn5                   |                   Syn6                   |
> --------------------------------------------------------------------------------------------------------------------------------------------------
>    Lambda / Metrics (%)  |         TPR        |      FDR        |         TPR        |      FDR       |         TPR        |      FDR        |
> --------------------------------------------------------------------------------------------------------------------------------------------------
>                  0.1                      |       98.0         |      94.3        |         90.0        |      93.4      |         99.2        |        92.3     |
>                  0.3                      |       93.7         |      87.9        |         84.2        |      88.9      |         96.9        |        86.7     |
>                  0.5                      |       99.0         |      43.1        |         88.3        |      50.6      |         99.6        |        31.7     |
>                    1                       |       66.3         |      40.5        |         73.2        |      23.7      |         90.5        |        15.4     |
>                    2                       |         0.0         |       0.0         |         25.4        |       4.1       |         67.1         |         3.6      |
>                    5                       |         0.0         |       0.0         |          7.5         |       2.7       |          7.6          |         2.5      |
>                   10                      |         0.0         |       0.0         |          0.0         |       0.0       |          0.0          |         0.0      |
> --------------------------------------------------------------------------------------------------------------------------------------------------
>
> A3: As can be seen in the GitHub code (anonymously published on https://github.com/iclr2018invase/INVASE), on testing data, we select the features whose selection probabilities are larger than 0.5. (see line 225 in INVASE-.py and line 274 in INVASE.py) We will clarify this in the revised manuscript.

---

> > ### Comment · AnonReviewer3 · 2018-11-21
> > **The feedback addressed my concerns**
> >
> > I would like to thank the authors for clarifying my concerns in details, especially for the first point. I think this is a straightforward idea that relaxes the need for a predefined k in L2X and has good performance. I have updated my score accordingly.

---

### Meta-Review · Area_Chair1 · 2018-12-13

**Confidence:** 3
**Recommendation:** Accept (Poster)

**Metareview:**

This manuscript proposes a new algorithm for instance-wise feature selection. To this end, the selection is achieved by combining three neural networks trained via an actor-critic methodology. The manuscript highlight that beyond prior work, this strategy enables the selection of a different number of features for each example. Encouraging results are provided on simulated data in comparison to related work, and on real data.

The reviewers and AC note issues with the evaluation of the proposed method. In particular, the evaluation of computer vision and natural language processing datasets may have further highlighted the performance of the proposed method. Further, while technically innovative, the approach is closely related to prior work (L2X) -- limiting the novelty.

The paper presents a promising new algorithm for training generative adversarial networks. The mathematical foundation for the method is novel and thoroughly motivated, the theoretical results are non-trivial and correct, and the experimental evaluation shows a substantial improvement over the state of the art.